# Molecular Mechanisms and Risk Factors Related to the Pathogenesis of Peyronie’s Disease

**DOI:** 10.3390/ijms241210133

**Published:** 2023-06-14

**Authors:** Yozo Mitsui, Fumito Yamabe, Shunsuke Hori, Masato Uetani, Hideyuki Kobayashi, Koichi Nagao, Koichi Nakajima

**Affiliations:** Department of Urology, Toho University Faculty of Medicine, Tokyo 143-8540, Japan; fumito.yamabe@med.toho-u.ac.jp (F.Y.); shunsuke.hori@med.toho-u.ac.jp (S.H.); masato.uetani@med.toho-u.ac.jp (M.U.); hideyukk@med.toho-u.ac.jp (H.K.); repro@med.toho-u.ac.jp (K.N.); koichin@med.toho-u.ac.jp (K.N.)

**Keywords:** Peyronie’s disease, pathogenesis, risk factors, molecular mechanisms, transforming growth factor-β1

## Abstract

Peyronie’s disease (PD) is a benign condition caused by plaque formation on the tunica albuginea of the penis. It is associated with penile pain, curvature, and shortening, and contributes to erectile dysfunction, which worsens patient quality of life. In recent years, research into understanding of the detailed mechanisms and risk factors involved in the development of PD has been increasing. In this review, the pathological mechanisms and several closely related signaling pathways, including TGF-β, WNT/β-catenin, Hedgehog, YAP/TAZ, MAPK, ROCK, and PI3K/AKT, are described. Findings regarding cross-talk among these pathways are then discussed to elucidate the complicated cascade behind tunica albuginea fibrosis. Finally, various risk factors including the genes involved in the development of PD are presented and their association with the disease summarized. The purpose of this review is to provide a better understanding regarding the involvement of risk factors in the molecular mechanisms associated with PD pathogenesis, as well as to provide insight into disease prevention and novel therapeutic interventions.

## 1. Introduction

Peyronie’s disease (PD) is a connective tissue disorder characterized by the formation of fibrous lesions or plaques in the tunica albuginea (TA) of the penis. The early stage is an active phase, with affected individuals experiencing erectile pain and penile curvature onset. However, not all cases present with penile pain at onset, with the incidence ranging from 20% to 70% of reported cases [1]. At 12–18 months following PD onset, the disease transitions to a painless chronic phase, with a gradual worsening of penile curvature in 20–50% of the cases [1,2,3,4], during which changes in the penis rarely show spontaneous reversal. Therefore, a severely deformed penis may cause sexual intercourse disorders, such as difficulty with insertion, pain during intercourse, body posture restrictions, and erectile dysfunction (ED), which can adversely affect psychological well-being and quality of life (QoL). Notably, it has been reported that PD patients have mental distress and a distorted image of themselves, and depression is suspected in a high proportion [5,6,7]. Furthermore, a large-scale survey of more than 8000 PD patients conducted in Sweden showed that they had increased risk of self-harm, as well as anxiety and depression [8], while a recent study showed that younger patients tended to suffer more psychological and physical symptoms, as well as penile pain in the chronic phase of PD [9].

A large variety of medical treatments have been suggested and are utilized for PD. Nonsurgical treatments for affected patients include oral, topical, intralesional, extracorporeal shockwave, and traction therapy. In terms of oral treatment options, results have shown that a low dose of tadalafil may slow the progression of penile curvature in patients in the acute phase [10], while intralesional therapy with collagenase clostridium histolyticum is reported to be capable of improving penile curvature [11]. Additionally, extracorporeal shockwave therapy (ESWT) can cause plaque damage, resulting in plaque resorption in some affected patients [12,13]. In cases of chronic PD, assistive devices (penile traction, vacuum devices, penile prostheses) may be effective for improving penile curvature [14,15]. When these treatments are ineffective, surgical treatment such as a grafting procedure is often attempted to correct penile curvature [16].

Although the exact etiology of PD remains unknown, evidence accumulated over the past two decades shows several molecular alterations and aberrant signaling pathways involved in the pathogenesis. In addition, epidemiological studies have indicated various risk factors possibly associated with its development, such as smoking, hypertension (HT), diabetes mellitus (DM), and older age [17,18]. Possible connections to certain genetic predispositions have also been proposed as being involved in PD development, including single nucleotide polymorphisms (SNPs) although the exact susceptibility factors have yet to be established [19,20,21,22]. The prevalence of PD has been reported to range from 0.6% to 20%, with wide variations among regions and ethnic groups although it is particularly low in Asians [22,23,24,25,26,27]. One reason for these differences may be the influence of certain genetic differences among regions and races. Such intrinsic and extrinsic risk factors, as well as comorbidities, may independently increase susceptibility to PD or function synergistically to increase the risk of disease development. A better understanding of both related molecular mechanisms and multifactorial factors associated with PD pathogenesis will have a major influence on treatment options as well as prevention strategies. The present literature review provides a summary of current findings regarding molecular mechanisms related to PD pathogenesis and the putative roles of several risk factors including the genetic factors associated with its development.

## 2. Pathogenesis of Plaque in PD Cases

The hypothesis posed by Devine et al. that PD is initiated by repetitive micro-trauma to the penis during intercourse has become widely accepted [28]. In one study, rats that received repeated local injections of chlorhexidine ethanol to produce repetitive microtrauma to the TA developed fibrous plaque in the penis [29]. For affected patients, the primary site of injury is thought to be the TA, with disruption of the inner and outer layers creating a milieu for inflammation, elastic fiber disruption, and extravascular blood accumulation [21]. This serves as a scaffold for fibrin and fibronectin accumulation, which is followed by infiltration of inflammation-associated cells that release various types of cytokines and subsequently show cell-type conversion, including tissue-resident fibroblasts and smooth muscle cells, into myofibroblasts characterized by α-smooth muscle actin (α-SMA) expression and collagen secretion. Transforming growth factor-β1 (TGF-β1) is a key factor for myofibroblast activation and a major player in fibrosis in all organs [19,20,21,30]. Myofibroblasts are a crucial element for the fibrotic process, as they cause excessive accumulation of collagen fibers, fibronectin, and other components of the extracellular matrix (ECM). An imbalance between matrix metalloproteinase (MMP), which removes collagen fibers, and its tissue inhibitor of metalloproteinase (TIMP) in myofibroblasts, along with the evasion of apoptosis, are thought to be key factors in the development of fibrotic diseases [21,31,32]. Interestingly, recent studies have revealed the involvement of persistent immunological features including mainly macrophages with fibrosis including PD [20,32,33,34,35,36,37]. Based on the latest reported findings, details regarding the mechanisms of fibrotic tissue remodeling and putative functions of each cell type during different stages of PD are presented below and in Figure 1.

### 2.1. Role of the Immune System in Inflammatory Response

In the tissue fibrosis process, innate and adaptive immune systems are both intricately involved in inflammatory events. Macrophages and monocytes are key immune-related cells related to the innate immune system, while T cells and B cells are involved in the adaptive immune system.

At the site of injury, platelets become activated, then release granules and stop blood loss first by forming an initial provisional matrix composed of fibrin and fibronectin, which has also been identified in PD [38]. Fibrin and fibronectin also function to alter leukocyte recruitment to the wound site and promote fibrosis [39]. The chemokine monocyte chemoattractant protein 1 (MCP-1), another factor that mediates recruitment and activation of macrophages and monocytes, is produced and secreted from monocytes/macrophages, fibroblasts, and vascular endothelial cells by stimulation of lipopolysaccharide and inflammatory cytokines upon tissue injury [40,41]. A gene expression profile study using PD-derived lesions demonstrated significantly elevated expression of MCP-1 in PD plaque over a normal TA condition [42], while Lin et al. confirmed higher levels of MCP-1 mRNA in PD than in non-PD cells [43]. Other factors involved in recruitment and stimulation of immune-related cells include pathogen-associated molecular patterns and damage-associated molecular patterns (DAMPs), which stimulate through various receptors on immune cells such as toll-like receptors (TLR) (mostly-2 and -4) [20,37,44]. DAMPs are released following stress or injury to various cells, including dead cells, proliferating neutrophils, macrophages, lymphocytes, natural killer cells, mesenchymal stem cells (MSCs), and resident cells [45,46,47]. In 2000, Mills et al. proposed a new classification for macrophages as either M1—pro-inflammatory— or M2—anti-inflammatory/profibrotic [48]. Thereafter, M2 macrophages were further classified into M2a, M2b, M2c, and M2d subtypes based on the secretion of distinct cytokines, the presence of certain cell surface proteins, gene expression profiles, and other biological activities, while they commonly express interleukin (IL)-10 [49,50].

There are different subtypes of T cells, each with unique regulatory effects on inflammation and fibrosis, including through cytokine release. Among these, helper T cells are a subpopulation of lymphocytes that express the CD4 antigen on the cell surface and are now mainly divided into Th1, Th2, and Th17 types [51]. Th1 cells differentiate generally in the presence of IL-12 and mainly produce interferon gamma (IFN-γ) after differentiation, whereas Th2 cells differentiate in the presence of IL-4 and continue to produce mainly IL-4 after differentiation, with each having an influence on macrophage activation [52]. In addition to granulocyte-macrophage colony-stimulating factor (GM-CSF) and tumor necrosis factor-alpha (TNF-α), the Th1 cell cytokine IFN-γ, activates polarization of M1 macrophages via various pathways, including the Janus kinase/signal transducer and activator of transcription (JAK/STAT) and nuclear factor (NF)-κB pathways [36,37,53]. The Th2 cytokine IL-4, similar to IL-13 and TGF-β1, activates M2 macrophages through STAT6 or interferon regulatory factor 4-mediated signaling [54,55,56], with this macrophage plasticity being known as M1/M2 polarization. M1 macrophages activated in this manner exhibit high levels of antigen-presenting activity and production of pro-inflammatory cytokines such as IL-6 and TNF-α, as well as the production of nitric oxide (NO) and reactive oxygen species (ROS) [36,37,55]. In contrast, M2 macrophages produce fewer of the inflammatory cytokines IL-6 and TNF-α. In particular, the M2c type is characterized by high expression levels of the anti-inflammatory cytokine IL-10 and profibrotic factor TGF-β1, thus contributing to tissue remodeling and fibrosis promotion [36,37,55,56,57].

B cells contribute to several different fibrotic diseases via a number of mechanisms, including direct cell–cell contact and the production of profibrotic and pro-inflammatory cytokines (e.g., IL-6 and TGF-β1) [58,59,60]. Furthermore, decreased IL-10 production by B cells is possibly a factor associated with fibrosis according to previous examinations of systemic sclerosis patients [60,61,62]. These studies also showed that increased IL-6 and reduced IL-10 levels in B cells are induced through the activation of the JAK/STAT and TLRs pathways.

M1/M2 macrophage phenotypes are interchangeable depending on the microenvironment, while an M1–M2 imbalance expressed by excessive M2 activity is associated with different types of fibrosis through excessive fibroblast activation [36,63,64]. Using a cisplatin-induced rat renal fibrosis model, Nakagawa et al. showed that the number of M1 macrophages begin to increase with the upregulation of M1-related cytokines (IFN-γ, TNF-α, IL-6) in the middle stage and the decrease in the late stage [65]. In contrast, M2 macrophages exhibit progressively increased numbers in the middle and late stages, accompanied by increased profibrotic factor TGF-β1 expression. This study also found increased numbers of CD4+ and CD8+ T cells in the late stage of the fibrotic process. Thus, M1 macrophages are involved in the relatively early stages of wound response and are thought to be replaced at later stages by specific repair populations of M2 macrophages via T–cell responses. Regarding PD, serum IL-6 levels were found to be significantly elevated in patients with PD in the acute phase as compared to healthy controls [66]. In contrast, a study that compared 91 PD patients with healthy controls showed that patients in the chronic stage had elevated serum levels of TGF-β1 and IFN-γ, and decreased levels of TNF-α, while IL-6 was undetectable [67]. In addition, a recent study by Cui et al. that used integrated bioinformatic analysis provided interesting findings regarding the association of PD with immune-related cells [68]. Research based on immune cell infiltration analysis demonstrated that various immune-related cells, such as T cells, B cells, macrophages, neutrophils, and plasma cells, are widely involved in fibrosis in PD patients. Additionally a number of genes were also identified, including inflammatory cytokines such as IL-6, as being important immune-related factors in PD and their correlations were shown. Notably, functional enrichment analysis showed that these genes have potential regulatory roles in several key biological processes, including the JAK/STAT pathway. Thus, complicated mechanisms that arise through M1–M2 macrophage interactions and T–cell responses may contribute to PD development.

### 2.2. Differentiation and Proliferation of Myofibroblasts in the Profibrotic Environment

Myofibroblasts, which function in tissue contraction and ECM secretion, have an important role during impaired healing. The profibrotic milieu, formed by the infiltration and activation of leukocytic tissue, as well as Th2 cell–M2 macrophage polarization, induces mesenchymal transition (conversion to myofibroblasts) in various cell types, such as tissue-resident fibroblasts, fibrocytes of different origin, smooth muscle and endothelial cells, and mesenchymal stem cells [21]. Various factors are involved in conversion to myofibroblasts, among which TGF-β1 plays a central role. Others include pro-inflammatory cytokines (e.g., TNF-α, IL-6, IL-1β), growth factors [e.g., platelet-derived growth factor (PDGF), basic fibroblast growth factor (bFGF)], myostatin, and ROS.

PDGF is primarily secreted by endothelial cells and macrophages, and is also released from platelets upon degranulation, with its production as well as expression of related receptors being maintained by TGF-β1 signaling [20]. Hamid et al. showed that pro–inflammatory macrophages promote the myofibroblast transformation of cardiac MSCs through the activation of PDGF and its receptor PDGFR-β [69], while another study using a rat PD model revealed significantly elevated expression levels of PDGF-α and PDGF-β as well as their receptors PDGFR-α and PDGFR-β [70]. Interestingly, a recent study that used single-cell RNA sequencing (scRNA-Seq) with samples from patients with Dupuytren’s disease (DD), widely known to be associated with PD, showed that the endothelium functions to promote an immune regulatory fibroblast phenotype through PDGF signaling during fibrosis development [71]. bFGF, one of the factors that is expressed in the early stage of tissue injury and that plays an important role in wound healing [72,73], not only functions as a fibroblast mitogen, but also promotes myofibroblast proliferation and migration after TGF-β1-induced mesenchymal transition [4]. It was also shown that production of bFGF by PD plaque-derived myofibroblasts is significantly greater than that of by normal TA myofibroblasts [74]. Furthermore, myostatin, a member of the TGF-β family, may promote fibrous tissue formation in muscle tissue by transforming fibroblasts into myofibroblasts to replace damaged muscle fibers [75]. A study of the effect of myostatin deficiency on alterations in left ventricular function after myocardial infarction showed that α-SMA activity was less intense in myostatin knockout than in wild-type mice, suggesting the possibility of fewer myofibroblasts [76]. Moreover, myostatin was found to be overexpressed in PD plaque and to stimulate myofibroblast generation and collagen expression in rat TA cells [77].

Accumulation of ROS is induced by hypoxia and produces oxidative stress, which has detrimental effects on various cellular components. ROS-induced myofibroblast transformation involves a pathway through ROS-mediated HIF-1α stabilization [78,79]. For example, HIF-1α was shown to induce abnormal fibroblast activity via the activation of the extracellular signal-related kinase signaling pathway in cutaneous wound healing [80]. Another study found that rats with PD-like lesions expressed high levels of HIF-1α and inducible NO synthase (iNOS) via the activation of NF-κB [70]. More recently, results of bioinformatic approaches identified a number of potential hypoxia-associated genes associated with PD [81].

### 2.3. Extracellular Matrix Deposition

Following an initial inflammatory event, fibrosis progression is facilitated by the accumulation of ECM components secreted by myofibroblasts, leading to fibrotic tissue remodeling. The predominant components include collagen, especially types I, III, and IV, in penile plaque, proteoglycans, fibronectin, and fibromodulin [19,82,83], with excessive accumulation caused by disruption of the complex interactions of various factors. 

MMPs, neutral proteases, play a central role in ECM degradation and when activated lead to ECM degradation during normal wound healing. Their activation is regulated by various factors, including TGF-β1, PDGFs, IL-1, TNF-α, and bFGF. For example, TGF-β1 activates collagenase MMPs (MMP-1, -8, and -13, which degrade types I, II, and III collagen) and gelatinase MMPs (MMP-2 and -9, which degrade types IV, V, and VII collagen, as well as gelatin and elastin) [84]. However, a previous study found that administration of TGF-β1 to a fibroblast cell line established from PD plaque did not alter MMP-1, -8, or -13 expressions and only mildly increased that of MMP-9, although IL-1β induced production of MMP-1, -3, -10, and -13 [85]. Another study compared samples from PD plaque and normal TA using immunostaining and also found no differences in MMP-2 or -9 expression between them [86]. Instead, this study confirmed that administration of TGF-β1 induced excessive accumulation of TIMP1-4, which suppresses the activity of MMPs, in fibroblast cell lines established from PD plaque [85]. TIMPs are mainly produced in macrophages, fibroblasts, and myofibroblasts, and their expression is also positively regulated by PDGF and bFGF. Plasminogen activator inhibitor type 1 (PAI-1), synthesized in various cell types including endothelial cells, fibroblasts, and macrophages, is also known to inhibit plasmin-mediated MMP activation and promote fibrosis [87]. Davila et al. found that both mRNA and protein levels of PAI-1 were significantly increased in human PD plaque samples and respective fibroblast cultures as compared to normal TA [88]. Therefore, excessive ECM accumulation during PD plaque development may be driven by altered MMP levels caused by the inhibitory effects of TIMP and PAI-1 on MMP activation.

### 2.4. Progression of Fibrosis in Environment with Continuous Inflammation

In normal wound healing, myofibroblasts contract via the increased expression of matrix protein receptor integrins to regulate excessively deposited ECM and then continue to support mechanical loads until collagen cross-linking for formation of striated scar tissue [39]. After the wound is repaired, myofibroblasts undergo apoptosis, and the repair response is terminated [82]. However, in fibrotic diseases including PD, tissue remodeling and fibroblast activation persist as chronic, uncontrolled processes, with structural changes in fibrotic tissue thought to play an important role in the formation of this chronic extracellular profibrotic milieu [89]. Progressive deposition of ECM proteins creates a hypoxic environment by increasing stiffness around the affected TA and inhibiting oxygen diffusion. Lack of oxygen, the terminal electron acceptor of the electron transport chain, increases ROS generation, further promoting cellular and tissue damage and myofibroblast activation [81,90]. ROS-induced damage then induces release of DAMPs from damaged cells, resulting in further continued recruitment and stimulation of macrophages and lymphocytes. Continuous recruitment and stimulation of immune-related cells is also maintained by fibronectin and MCP-1 released from TGF-β1-activated myofibroblasts [91]. Thus, structural changes in fibrous tissue due to excessive ECM deposition may lead to sustained myofibroblast activation, contributing to continuation of the profibrotic milieu.

The effect of ESWT on PD in the early phase is thought to be related to direct destruction of the penile plaque by shockwaves, as well as plaque lysis caused by the activation of macrophages induced by an inflammatory response to heat generated by the treatment [92]. However, as noted above, macrophage activation is a major factor that promotes fibrosis, and the clinical significance of ESWT remains questionable. A recently published systematic review and meta-analysis indicates that ESWT may reduce plaque size, although it fails to improve penile curvature or pain in PD patients [13].

Evasion of myofibroblast apoptosis is also critically involved in maintaining a chronic extracellular fibrous milieu. Various types of programmed cell death, including apoptosis, are known to be closely associated with organ fibrosis [93]. Apoptotic pathways include intrinsic pathways activated by cellular stress and DNA damage–as among other causes–and extrinsic pathways activated via the detection of extracellular death signals from other cells. The intrinsic apoptotic pathway is tightly regulated by a balance of the activities of Bcl-2 family proteins, consisting of pro–apoptotic (Bax, Bak, Bad, Bid, Puma, Bim, Noxa) and anti–apoptotic (Bcl-2, Bcl-xL, Bcl-w, Mcl-1) proteins. Furthermore, both intrinsic and extrinsic apoptotic pathways ultimately depend on the protease activity of the initiator (e.g., caspase-2, -8, -9, -10, -12) and executioner (e.g., caspase-3, -6, -7). Interestingly, Zorba et al. found high levels of Bcl-2 expression and decreased levels of caspase-3 and -8 expression in PD plaque, indicating apoptosis suppression [94]. In contrast, Loreto et al. showed that PD tissues had a lower expression of Bcl-2 but an overexpression of Bax and caspase-3 and -9 as compared to normal TA [95]. They also reported that a tumor necrosis factor-related apoptosis-inducing ligand and its receptor DR5 were overexpressed in PD tissues, suggesting that apoptosis may be induced by intrinsic and extrinsic pathways [96]. These different results may be partially explained by a new concept of apoptosis evasion in myofibroblasts termed “mitochondrial priming” [97], which refers to the proximity of mitochondria to the apoptotic threshold and which is determined by the relative expression of pro–apoptotic and pro–survival members of the Bcl-2 family of proteins. Myofibroblasts are “primed” and thus ready to die at the time of conversion from fibroblasts and become dependent on pro–survival proteins to sequester pro–apoptotic proteins and ensure survival. Therefore, various types of apoptosis-associated proteins are expressed at different ratios in the mitochondria of myofibroblasts that continue to survive while avoiding apoptosis, which might be a factor contributing to differences in these results. Additionally, epigenetic modifications may lead to sustained myofibroblast activation and consequent fibrosis, the details of which will be discussed later.

PD plaques formed in this way evolve toward calcification in 20–25% of affected cases [98]. PDGF functions as an osteoblast recruiter and contributes to calcification and ossification of PD plaques [70]. Following the initial inflammatory phase, the disease moves to the second phase (chronic), during which time the disease stabilizes. The absence of pain and inflammation has long been thought to be a hallmark of the chronic phase. However, a recent study by Milenkovic et al. that used immunohistochemistry and scRNA-Seq indicated the possibility of a sustained inflammatory reaction even in the chronic PD stage [99]. Such findings showing inflammation in PD may pave the way for the development of new treatments for patients in the chronic phase. A brief summary of the key molecules and their roles in the pathophysiology of PD is presented in Table 1.

## 3. Core Signaling Pathways and Their Crosstalk Involved in Development of Peyronie’s Disease

TGF-β, a central regulator of tissue inflammation, repair, remodeling, and fibrosis, has three isoforms: TGF-β1, TGF-β2, and TGF-β3 [100]. TGF-β is stored in a latent form in most normal tissues, while upon tissue injury, de novo synthesis of TGF-β isoforms and latent stores of TGF-β are activated. In particular, deregulation of TGF-β1 activity and thereby activation of TGF-β signaling have been strongly linked to the pathogenesis of PD, as described above. Indeed, activation of TGF-β signaling through the subtunical injection of TGF-β1 in rat and rabbit models has been demonstrated to induce PD-like plaque formation in the penis [101,102]. In addition, there are other critical target molecules and signaling cell pathways related to fibrosis, which develops when these interact around the TGF-β signaling pathway. Those potentially associated with PD include the Wingless/int1 (WNT)/β-catenin, Hedgehog, Yes-associated protein 1(YAP)/transcriptional coactivator with PDZ-binding motif (TAZ), mitogen-activated protein kinase (MAPK), RAS homologue gene family (RHO)-associated kinase (ROCK), and phosphoinositide 3-kinase (PI3K)/RAC-α serine/threonine protein kinase (AKT) signal pathways. In this section, the detailed mechanisms of TGF-β1 synthesis and activation in the pro–fibrotic environment are reviewed, along with interactions of the TGF-β signaling pathway involved with the other core signaling pathways in PD development.

### 3.1. TGF-β1 Synthesis and Aberrant Activation of Latent TGF-β1

TGF-β1 is produced by platelets, fibroblasts, and all leukocyte lineage cells [103]. Among these, M2 macrophages play an important role in production and activation of TGF-β1 in fibrotic diseases including PD. PAI-1, which promotes fibrosis through inhibition of MMP activation, is also a factor that promotes TGF-β1 synthesis, while its expression is enhanced by TGF-β1 [104]. Enhanced serotonin signaling mediated by 5-hydroxytryptamine receptors resulting in increased transcription of TGF-β1 has been identified in some fibrotic diseases [105,106]. TGF-β1 is synthesized in a linked state with latency-associated polypeptide (LAP), then forms a complex with latent TGF-β-binding protein (LTBP) or a leucine-rich repeat containing (LRRC) protein, and is subsequently released outside the cell [107]. Thus, synthesized TGF-β1 (as well as the remaining variants) is secreted extracellularly as biologically inactive latent TGF-β1. Released latent TGF-β1 is anchored using LTBP to ECM and LRRC protein molecules on the cell surface to maintain its inactivated state. Several factors, including integrins, thrombospondin 1, ROS, and several proteases (plasmin, cathepsin D, MMP-2, MMP-9), activate latent TGF-β1 by inducing LAP degradation, denaturation, or conformational changes [84,108,109]. Among them, the expression of thrombospondin 1 is known to be promoted by the renin-angiotensin system (RAS) [110], which is involved in the regulation of fibrosis, inflammation, and oxidative stress, and plays an important role in aging and age-related chronic diseases [111]. Notably, inhibition of this pathway with an angiotensin II receptor blocker has been shown to attenuate penile fibrosis in a rat cavernous nerve injury model [112,113]. Another study found that protein the expressions of angiotensin-converting enzyme, AT1 receptor, and Mas receptor were upregulated in aged–rat penile tissue characterized by increased collagen deposition [114]. The putative mechanisms of TGF-β1 activation during the development of PD based on these findings are shown in Figure 2.

### 3.2. Aberrant TGF-β Signaling Pathway in the Pathogenesis of Peyronie’s Disease

TGF-β signaling includes SMAD-mediated canonical and non-canonical TGF-β signaling pathways. Canonical TGF-β signaling is initiated when a TGF-β ligand (TGF-β1, 2, 3) binds to the TGF-β type II receptor (TGF-βRII). TGF-βRII subsequently promotes phosphorylation of the TGF-β type I receptor (TGF-βRI), which signals through the phosphorylation of SMAD2/SMAD3, triggering a cascade response. Phosphorylated SMAD2/SMAD3 proteins in a complex with SMAD4 then translocate to the nucleus, where the complex binds to specific DNA regions, SMAD-binding elements, to induce the transcription of profibrotic genes (αSMA, collagen, PAI-1, TGF, TGF-βR1, MMPs, fibronectin, etc.). Following TGF-β1 stimulation, PD plaque-derived fibroblasts show faster nuclear translocation of SMAD2/SMAD3, and higher expression levels of SMAD3 and SMAD4 as compared to normal cells [115]. In addition to excessive TGF-β1 activation, several factors may cause such aberrant TGF-β/SMAD signaling activation and promote PD fibrosis.

The TGF-β/SMAD signaling pathway is regulated by feedback inhibition to control cellular homeostasis. SMAD7, an inhibitory SMAD, provides competition for TGF-βRI and inhibits the TGF-β/SMAD signaling pathway by blocking SMAD2 phosphorylation and activation [116]. Magee et al. assessed gene expression differences between PD plaque tissue and normal TA using DNA microarray findings and reported that SMAD7 was the most significantly downregulated gene in PD plaque tissue [42], suggesting that downregulation of SMAD7 expression activates the TGF-β/SMAD signaling pathway in PD. These observations were supported by other later studies. Specifically, Choi et al. found that human PD plaque-derived fibroblasts with forced expression of the SMAD7 gene inhibited the phosphorylation and nuclear translocation of SMAD2 and SMAD3, and suppressed ECM protein production by TGF-β1 [117]. They also confirmed that SMAD7 induces apoptosis and suppresses cell cycle entry in PD plaque-derived fibroblasts. Wang et al. reported that the anti–fibrotic effect of rat bone marrow MSC treatment in PD rat models was associated with increased SMAD7 expression but that this was offset by knockdown of Smad7 expression [118]. Another study suggested that connective tissue growth factor (CTGF) promotes TGF-β1 activity by reducing SMAD7 expression and activating SMAD2 phosphorylation [119]. CTGF, itself also factor whose expression is promoted by TGF-β1, contributes to the maintenance of the fibrotic process [120]. Although fibroblasts derived from PD patients did not show upregulation of CTGF [43], a pilot study of prostacyclin 12 analog, a drug for PD patients that suppresses CTGF expression, noted that penile curvature was improved in approximately 30% of examined patients [121]. Insulin-growth factor 1 (IGF-1), a major mediator of growth hormone signaling, is also associated with various fibrotic diseases. Sarenac et al. showed that its administration to human keratinocytes inhibited transition of SMAD3 to nucleation, increased Smad7 expression, and inhibited the TGF-β/SMAD pathway leading to fibrosis [122]. Interestingly, PD plaque shows a significantly lower expression of all IGF-1 isoforms than does normal TA, suggesting that the suppression of the TGF-β/SMAD pathway by IGF-1 may be reduced in PD [123].

Apart from SMAD signaling, TGF-β ligands can activate multiple alternative pathways, commonly referred to as “non-canonical” TGF-β signaling pathways that are relevant to fibrosis pathogenesis. Activation of the MAPK signaling pathways mediated by extracellular signal-regulated kinase (ERK), JUN N-terminal kinase (JNK), and p38 has been implicated in the SMAD-independent production of a mediator that promotes development of fibrotic disease. An in vivo experiment performed by Chatzifrangkeskou et al. showed that ERK1/2 acts directly on induction of CTGF expression to mediate myocardial fibrosis and left ventricular dysfunction [124]. Activated by profibrotic factors such as TGF-β1, JNK is subsequently translocated into the nucleus where it regulates transcription of genes involved in inflammatory and fibrotic responses through the phosphorylation of c-Jun and SMAD3, leading to kidney inflammation and fibrosis [125]. In the liver, activation of the TGF-β1-mediated p38 MAPK signaling pathway may be associated with liver fibrosis via the accumulation of ECM through transcriptional activity of α-SMA and collagen [126]. As for penile tissues, studies of animal models showed JNK involvement in corpus cavernosum apoptosis in the acute phase after corpus cavernosum nerve crush injury and found that the activation of the p38 MAPK pathway induces penile vascular dysfunction in association with diabetes [127,128].

PI3K/AKT signaling is also induced by TGF-β1, and their dysregulation is associated with various degenerative conditions, including fibrotic diseases. Inhibition of this signaling pathway suppresses TGF-β-induced differentiation of human lung or cardiac fibroblasts into desmoplastic myofibroblasts for escape from the fibrotic process [129,130]. A profibrotic mechanism of AKT activated by TGF-β1 stimulation is inactivation of the inhibitory effect of nuclear receptor subfamily 4 group A member 1 (NR4A1), which inhibits profibrotic gene expression in the nucleus [37]. Another study found that NR4A1 agonists prevented NR4A1 inactivation and exerted antifibrotic effects in many organs of experimental fibrosis models [131]. Furthermore, Jung et al. showed that the inhibition of PI3K/AKT signaling by PI3K inhibitors suppresses fibrotic responses, such as cell proliferation and collagen synthesis, in PD-derived primary fibroblasts [132].

### 3.3. Another Core Signaling Pathway and Crosstalk in Peyronie’s Disease

The WNT/β-catenin signaling pathway is essential for organ development and generation although its pathological activation is associated with several fibrotic diseases. Thus, targeted inhibition of this pathway results in antifibrotic effects in a variety of organs [133,134,135]. Under inactivation of WNT signaling, β-catenin is normally phosphorylated in NH2-terminal residues, in which a glycogen synthase kinase 3β (GSK-3β) consensus motif is present, with the aid of a scaffolding complex composed of axin and adenomatous polyposis proteins. Phosphorylated β-catenin is then degraded by the ubiquitin–proteasome system. On the other hand, with the activation of WNT signaling, loss of GSK-3β function prevents phosphorylation of β-catenin. Subsequently, the unphosphorylated form of β-catenin accumulates in the cytoplasm, enters the nucleus, and finally activates WNT target genes as a transcriptional activator with help from transcription factor (TCF)/lymphoid enhancer-binding factor (LEF) family proteins. Previous studies have shown that β-catenin is significantly upregulated in cells derived from PD plaque as compared to normal TA tissue [136]. Ten Dam et al. performed immunostaining of PD plaque and normal TA tissues, and found significant upregulation of several WNT ligands in the PD samples, including WNT2, 4, 5a, and 7b, as well as β-catenin [137]. These results suggest that the activation of the WNT/β-catenin signaling pathway is associated with PD development.

The WNT/β-catenin signaling pathway activated in fibrotic diseases involves extensive crosstalk between the TGF-β and Hedgehog signaling pathways. The dickkopf-related protein 1 (DKK1) inhibits WNT/β-catenin signaling by forming a complex with low-density lipoprotein receptor-related proteins, which inhibits interactions between WNT and the Frizzled receptor. Secreted frizzled-related protein1 (SFRP1) is also a WNT/β-catenin signaling antagonist, which, unlike DKK1, directly binds to WNT and inhibits its binding to the Frizzled receptor. TGF-β1 also inhibits the transcription of DKK1 and SFRP1 in a p38-dependent manner and induces epigenetic silencing through DNA methylation, leading to further activation of the WNT/β-catenin signaling pathway [138,139,140]. Hedgehog signaling stimulates fibroblast-to-myofibroblast transition and promotes fibrosis. Indeed, pharmacological or genetic inactivation of hedgehog signaling has been shown to ameliorate experimental fibrosis in several mouse models of organ fibrosis [141,142,143]. Activation of hedgehog signaling is caused, at least in part, by TGF-β1 signaling, which induces transcription of the profibrotic Hedgehog transcription factor GLI family zinc finger 2 (GLI2) [144]. In addition, activation of Hedgehog signaling may upregulate Wnt-2b and Wnt-5a, shown to be overexpressed in PD, and may also activate WNT/β-catenin signaling [137,145].

The YAP/TAZ signaling pathway is regulated by both the HIPPO signaling pathway and a mechanically regulated HIPPO-independent mechanism [146]. In addition to HIPPO control, deregulation of YAP and TAZ driven by convergence of different activating signals, such as RHOA/ROCK, WNT/β-catenin, and TGF-β, is associated with fibrosis in various organs including PD [147]. Increased ECM stiffness due to fibrotic tissue remodeling is sensed by αVβ integrins, which are upregulated upon injury and activate intracellular signaling pathways involving RHOA/ROCK signaling. Subsequently, formation of F-actin stress fibers and translocation of YAP/TAZ to the nucleus are activated, where these proteins promote profibrotic gene expression through activation of the TEAD family member transcription enhancer factor (TEF). Furthermore, RHOA/ROCK signaling activates AKT, which may activate p300 in the nucleus via AKT-induced phosphorylation, resulting in further enhanced transcription of profibrotic genes [148]. Gene expression comparisons between normal TA and PD plaque performed by Qian et al. showed increased expression of β integrin and RhOA12 in PD [149]. Additionally, YAP1 expression was found to be upregulated in PD plaque samples as compared to normal TA tissues in a previous immunohistochemical study [137]. Milenkovic et al. reported that simvastatin and an RHO kinase inhibitor prevented myofibroblast transformation in PD-derived fibroblasts through the inhibition of YAP/TAZ nuclear translocation, and reduced CTGF and collagen expression [150]. Moreover, integrin-linked kinase (ILK) is involved in integrin-mediated signaling, promotes the inhibition of phosphorylation of myosin phosphatase target subunit 1, and inactivates Merlin, a key upstream regulator of the Hippo pathway, which activates YAP/TAZ in a large–tumor suppressor kinase (LATS)-1/2-dependent manner [151]; meanwhile, the ILK-mediated deregulation of the Hippo-YAP/TAZ pathway plays an important role in formation of keloids and hypertrophic scars [152]. In human corpus cavernosum smooth muscle cells, ILK may be a key regulator of biological functions, such as cell attachment, spreading, migration, and microfilament dynamics [153].

YAP and TAZ also bind phosphorylated SMAD2/SMAD3 and SMAD4 complexes formed in the nucleus through the activation of the TGF-β1 pathway. Subsequently, transcription of Serpin family E member 1 (SERPINE1), also known as PAI-1, is induced, which leads to excessive accumulation of ECM and contributes to the development of fibrotic diseases including PD [154]. Activation of YAP/TAZ signaling via RHOA/ROCK signaling is also induced by TGF-β1 as one of the non-canonical TGF-β signaling pathways. Jiang et al. showed that suppression of both TGF-β1-mediated SMAD and the RHOA/ROCK signaling pathway mediated by estrogen may suppress collagen secretion and self-contraction of TA-derived myofibroblasts [155].

Therefore, parallel non-regulated activity of several core pathways, with TGF-β signaling pathway as the main axis, that simultaneously leads to altered signaling crosstalk can conceivably result in the development of PD via aberrant tissue repair and fibrosis in TA tissues. A diagram showing the proposed possible core pathways involved in development of PD and their interrelationships is presented in Figure 3, while an overview of those is shown in Table 2.

## 4. Risk Factors for Pathogenesis of Peyronie’s Disease

Multifactorial diseases, characterized by inter-individual variations in etiology, onset age, and penetrance, are relatively common, and arise from the combined activities of genetic and environmental factors [156]. Multifactorial diseases develop when quantitative fluctuations of biological risk factors exceed certain thresholds due to the interactions of mutations in causative genes, such as polygenes and major genes, as well as the influence of environmental factors [157]. Accumulating evidence suggests that various genetic and environmental factors are involved in several molecular alterations that contribute to PD pathogenesis (Figure 4). PD is thus considered to be a multifactorial disease, and the integrated understanding of the complex mechanisms underlying gene-environment interactions associated with PD risk and progression could improve various aspects of patient care. In this section, genetic and precipitating risk factors associated with PD based on their speculated role in its pathogenesis are summarized. The following section gives a summary of risk factors, and their predicted impact on PD development is presented in Table 3.

### 4.1. Intrinsic Risk Factors

Genetic factors have been reported as being representative intrinsic factors possibly associated with PD development. Several studies have revealed the involvement of chromosomal aberrations, gene mutations, and epigenetic modifications in this disease. Although no genome-wide association study (GWAS) studies to date have specifically examined genetic susceptibility to PD, other reported findings provide several possibly associated candidate genes. Other known intrinsic risk factors include aging and low testosterone level, while associations of the ABO blood group with congenital penile curvature (CPC) and PD have also been reported [177,184].

#### 4.1.1. Genetics of Peyronie’s Disease

A pedigree analysis of three families affected by both PD and DD provided the first clue of genetic susceptibility to PD, as an autosomal dominant mode inheritance with incomplete penetrance was identified in all three, with one family exhibiting three generations of father to-son transmission [185]. Chromosomal abnormalities associated with PD have also been shown using fibroblast culture models. Somers found that karyotypic abnormalities in PD plaque-derived fibroblasts included duplication of chromosome 7 and 8, deletion of chromosome Y, and reciprocal translocations of 46XY, t(11;12)(q11,p11) and 46XY, t(1;5)(q25;q11), as well as inversion of 46XY, inv(7)(p22q36) [158]. These chromosomal abnormalities have been proposed to be acquired changes because they have been found to occur only in PD-derived fibroblasts and not in other tissue-derived cells. Mulhall et al. detected chromosomal aberrations involving chromosomes 7, 8, 17, 18, Y, and X in PD plaque-derived fibroblasts and confirmed that such fibroblasts showed earlier chromosomal instability than did those derived from normal TA [159].

A relationship between heritable SNPs and PD that can affect gene expression has been shown by several studies. We previously reported that elevated TGF-β1 level and activation of TGF-β signaling are the most important factors for development of PD. It is speculated that elevated TGFβ-1 level is partly due to the presence of SNPs within the *TGF*-*β1* gene. The SNP rs1800471 (p.A35P), which results in substitution of arginine to proline at position 25 in the TGF-β1 protein, might be a PD-associated SNP, as it has been found to occur at a significantly higher frequency in PD patients than in healthy subjects [160]. A study by Dolmans et al., which assessed allele frequencies of SNPs associated with the WNT/β-catenin pathway, found that rs4730775 in the *WNT2* gene locus on chromosome 7 was associated with PD pathogenesis [161]. As noted previously, WNT2 is one of the WNT ligands significantly upregulated in PD [137]. Furthermore, intriguing results were recently obtained by whole genome sequencing using blood samples from three PD patients with DD performed by Dullea et al. [162], as protein–encoding mutations in the *ALMS1* gene including the SNPs rs45501594 (p.T3445S), rs34071195 (p.K3435E), and rs41291187 (p.H624R) were present in all three. Compound heterozygous mutations of the *ALMS1* gene are known to cause Alström syndrome, a rare autosomal recessive disorder related to a variety of symptoms, such as cone–rod retinal dystrophy, sensorineural hearing loss, obesity, insulin resistance, DM, hypertriglyceridemia, and dilated cardiomyopathy [186]. Notably, ALMS1-deficient fibroblasts were shown to acquire apoptosis resistance and overexpress ECM components [187]. Based on reports that an *ALMS1* mutation is associated with aberrant TGF-β signaling [163,164], Dullea et al. speculated that such a mutation leads to increased TGF-β level, resulting in fibrosis and the PD phenotype [162].

Epigenetic modifications, such as DNA methylation, histone acetylation/deacetylation, and non–coding RNAs [e.g., microRNAs (miRNAs)], have a crucial role in fibrotic disease pathogenesis [37,167]. Such epigenetic changes may contribute to the development of fibrotic diseases by leading to sustained myofibroblast activation [165,188]. Specifically, histone deacetylases (HDACs) promote deacetylation of histones, major constituents of the chromatin structure, through TGF-β signaling and promote fibrosis via downregulation of the anti–fibrotic gene *PPARGC1A* [165]. Interestingly, Kang et al. showed that HDACs were more highly expressed in PD plaque samples as compared to normal TA. They also found that the silencing of HDAC7 attenuated TGF-β1-induced profibrinolytic responses in PD plaque–derived fibroblasts [166]. Among the various miRNAs associated with fibrosis, a recent study showed that the miRNA-29 family (miR-29a, b, c) has a key role in the process of multi–organ fibrosis [167]. Members of this family may be closely associated with regulating the expression of multiple cytokines; key fibrotic pathways such as TGF-β1/SMAD, PI3K/AKT, and DNA methylation; and the epithelial–mesenchymal transition of fibroblasts [167]. Dos Santos and colleagues also recently found that miR-29b was significantly downregulated in fibrous plaque, TA, and corpus cavernosum samples from PD patients as compared with those from controls [168]. In addition, results from studies that investigated the role of immunological contributors in PD suggest that several miRNAs have regulatory roles in the immune system of affected individuals [68].

#### 4.1.2. Aging

Aging, characterized by progressive loss of physiological integrity, is a major risk factor for human pathologies such as DM, cardiovascular disorders, and neurodegenerative diseases, while fibrosis is also considered to be a universal age-related pathological process in various organs. The prevalence of PD increases in the third decade and peaks in the fifth, suggesting aging as risk factor for this disease [18,189]. A decline in alpha-1-antitrypsin, a key protease inhibitor that regulates tissue degradation, was revealed to be an age-related phenomenon in PD-affected patients based on comparisons with age-matched healthy controls [169]. Another possible reason for the increased prevalence of PD with aging is unique microtrauma effects on the penis that differ as compared to those in young men because erections are less rigid [170]. Furthermore, in older men, the tissue becomes less elastic, and the deforming force of intercourse can cause bending, leading to minor trauma [171]. Aging has numerous functionally interrelated hallmarks, including stem cell depletion, genomic instability, telomere shortening, epigenetic changes, and loss of protein homeostasis [190]. Thus, genetic alterations and epigenetic modifications associated with PD noted in the previous section may also be induced by aging.

#### 4.1.3. Testosterone Level

Testosterone deficiency (TD) has been found to cause changes in TA and affect the collagen metabolism abnormalities associated with PD [191,192]. Subsequent clinical studies have also indicated that TD may influence PD etiology and severity [172,173,174,175]. However, any association between testosterone and PD remains controversial, as various presented findings have denied such as relationship [193,194,195]. In prostate cancer cells, the androgen receptor (AR), activated by testosterone binding, may attenuate TGF-β signaling by suppressing TGF-βRII expression [176]. AR is also expressed in penile tissues [196], and we speculate that decreased serum testosterone levels may induce TGF-βRII expression through decreased activation of AR, which activates the TGF-β signaling associated with PD development.

#### 4.1.4. ABO Blood Type

We recently compared 202 Japanese PD patients with 846 non-PD male patients and found that ABO blood type may be associated with PD risk [177]. Interestingly, as compared with individuals with blood type B, those with type O are at greater risk of PD, with an odds ratio of 2.018, and also have higher serum levels of the profibrotic cytokines TNF-α and IL-6 as compared to others [178,179]. TGF-βR1 is located on chromosome 9q22 near the ABO locus (9q34), and thus a relatively high rate of recombination between TGF-β1R1 and ABO-type genes can be assumed, as there may be an increased likelihood that these will be inherited together with ABO.

#### 4.1.5. Congenital Penile Curvature

CPC is a genetically inherited condition characterized by penis curvature that is present at birth without obvious organic pathological changes in the organ. Thus, CPC is considered to have a completely different pathology from PD, an acquired condition that causes obvious pathological changes in the TA. Recently, Paulis et al. conducted a retrospective analysis of a clinical database from a single andrology clinic and found a history of CPC in 73 of 519 patients with PD (14.1%), whereas only 15 (0.7%) of 2166 in a comparator population without PD had such results [184]. These authors speculated that the presence of CPC is a risk factor for the subsequent development of PD, although the mechanisms underlying the association between the two diseases remain unclear. It is possible that penile curvature causes increased damage to the penis during intercourse, facilitating accumulation of continuous TA microtrauma that triggers PD.

### 4.2. Extrinsic Risk Factors

Extrinsic risk factors are those have adverse effects on the body from outside sources, such as chemical and physical contaminants found in food, or appear as contaminants in the environment. Smoking, alcohol consumption, and perineal and penile trauma are possible extrinsic risk factors associated with PD.

#### 4.2.1. Smoking

Numerous studies have shown that smoking is strongly associated with PD [18,197,198,199]. Interestingly, the prevalence of PD increases with the number of cigarettes smoked per day as well as habit duration, with a particularly high prevalence reported in individuals who have smoked for more than five years [198,199]. An investigation of the systemic effects of chronic smoking on skin architecture confirmed extensive remodeling of the dermal elastic fiber system [200]. Liu et al. showed that cigarette smoke-induced fibrosis in human kidney cell lines was related to the inhibition of the ROS pathway and the induction of expression of the profibrotic factors CTGF and PAI-1 [180]. Furthermore, cigarette smoke was confirmed to induce epithelial-to-mesenchymal transition in non–small–cell lung cancer through HDAC-mediated downregulation of E-cadherin, a prognostic factor for lung cancer in smokers [181]. Smoking-induced ROS and epigenetic changes may also be involved in PD development.

#### 4.2.2. Alcohol

Alcohol consumption is a well-known risk factor for the development of several diseases in various organs, including fibrosis. For example, DM, dyslipidemia, and cardiovascular disorders, known PD-related comorbidities, have been shown to be strongly associated with alcohol consumption. A previous case–control study found that alcohol consumption was more frequent in PD cases as compared with a control group, although there was no difference related to the duration of alcohol consumption or types of beverages consumed [197]. Rat studies have also shown that long-term heavy alcohol consumption increases the expression of fibrotic cytokines such as TIMP-1 and SMAD3, leading to kidney tissue fibrosis [182]. Additionally, a study that investigated the early effects of alcohol intoxication on liver fibrosis in adolescents reported significantly elevated serum MMP-9 and TIMP-1 concentrations during intoxication [183].

#### 4.2.3. Perineal and Penile Trauma

As noted previously, PD initiation by penile microtrauma is the most widely accepted theory and a number of studies have shown that a history of penile trauma may be associated with PD [18,28,197,201]. Bjekic et al. reported that about a quarter of their examined PD patients had incidental genital perineal injuries, including iatrogenic injury caused by a therapeutic procedure such as catheterization or cystoscopy [202]. In addition, a history of penile trauma during intercourse has been reported in 21–40% of examined PD patients [28,201]. Such damage to the penis may cause detachment of septal fibers with extravasation of blood into the intraluminal space, leading to infiltration of inflammatory cells that trigger PD development.

### 4.3. Comorbidities

DM, HT, dyslipidemia, and obesity are widely recognized as comorbidities associated with PD. Although these and other comorbidities are often included in the category of intrinsic risk factors in a broad sense, the present review discusses them separately.

#### Hypertension, Diabetes Mellitus, Dyslipidemia, and Obesity

DM, cardiovascular disorders including HT, dyslipidemia, and obesity are considered as so-called lifestyle diseases. Many of these chronic conditions develop in middle or later life and, similarly to PD, are considered to be multifactorial and their association with PD has been shown by previous epidemiological studies [17,18,197]. DM, HT, and dyslipidemia are risk factors for systemic vascular disease, and may induce fibrosis in systemic organs through several mechanisms, such as vascular endothelial cell damage or ROS production. A high-fat, high-cholesterol diet leads to the development of steatohepatitis and severe fibrosis, which have been reported to be exacerbated by HT through NF-κB and MAPK pathways in rat models [202,203]. Furthermore, MMPs, TIMPs, and TGF-β1 have been identified as common proteins in signaling pathways involved in the fibroproliferative processes of PD and DM, indicating a common link [17].

## 5. Other Conditions Possibly Associated with Peyronie’s Disease

A major and frequent complication noted in PD patients is ED, while a high proportion are known to have DD. Interestingly, recent GWASs for these two diseases suggest genetic commonalities between them and PD. In this section, the pathophysiological mechanisms and genetic similarities that may be associated with ED and/or DD in cases of PD are discussed.

### 5.1. Erectile Dysfunction

ED is an important symptom of PD, as it impairs QoL, with previous studies showing a prevalence in more than 50% of PD patients [22]. This may be a symptom as well as cause of PD because of stronger trauma due to a semi–rigid erection or delayed ejaculation. Furthermore, complex factors are associated with development of ED, as those such as aging, DM, HT, and dyslipidemia associated with PD development are also risk factors for ED. Recently, two GWASs identified loci adjacent to single-minded homolog 1, an obesity-related gene, that may confer higher ED risk [204,205]. These findings have led to renewed interest in understanding the genetic basis of ED and PD [206].

### 5.2. Dupuytren’s Disease

PD is known to frequently occur together with DD, which is characterized by progressive fibrosis of the palmar and finger fascia [17,149,197]. Both PD and DD are focal manifestations of a systemic fibroproliferative pathological process, and share common risk factors such as aging, smoking, alcohol use, and diabetes [207]. In addition, several commonalities in their genetics and etiology have been reported [20,21,22,150,162]. A GWAS performed by Dolmans et al. identified nine chromosomal loci associated with susceptibility to DD [208], with 17 additional variants subsequently reported, bringing the current known total to 26 [209,210]. Notably, many of these include genes involved in the WNT/β-catenin signaling pathway that have been implicated in PD.

## 6. Conclusions

To date, several basic and clinical studies have revealed the precise detailed mechanisms involved in development of PD. Maintenance of the inflammatory environment by macrophages and T cells seems to provide an environment favorable for PD-related plaque formation. In short, a continuous inflammatory environment beginning after penile injury continues to support myofibroblast induction and activation until plaque formation. Furthermore, recent reports suggest that inflammation-associated immune system cells are present within lesions even in chronic-phase PD cases, in which inflammation is thought to have disappeared. Aberrant tissue repair and fibrosis of TA tissue under these circumstances are promoted by the interrelated effects of abnormalities in the TGF-β, WNT/β-catenin, Hedgehog, YAP/TAZ, p38 MAPK, ROCK, and PI3K/AKT signaling pathways. PD is manifests in individuals who cross the threshold for disease development, with the influence of various endogenous and extrinsic factors providing pleiotropic stimulation to pathways related with its development. Several reports have elucidated associated genetic abnormalities, and identification of PD susceptibility genes using GWAS is expected in the near future. A better understanding of the molecular mechanisms involved in the pathogenesis and how risk translates into disease will aid in the earlier recognition of PD and its prevention while also providing insight related to the development of adjunct and novel therapeutic options for affected individuals. Although the present review has several limitations, including the non-uniformity of the referenced literature (a mixture of clinical and basic studies) and the subjective method used for selection of those studies, it is our hope that the information presented herein will contribute to a deeper understanding of PD.

## Figures and Tables

**Figure 1 ijms-24-10133-f001:**
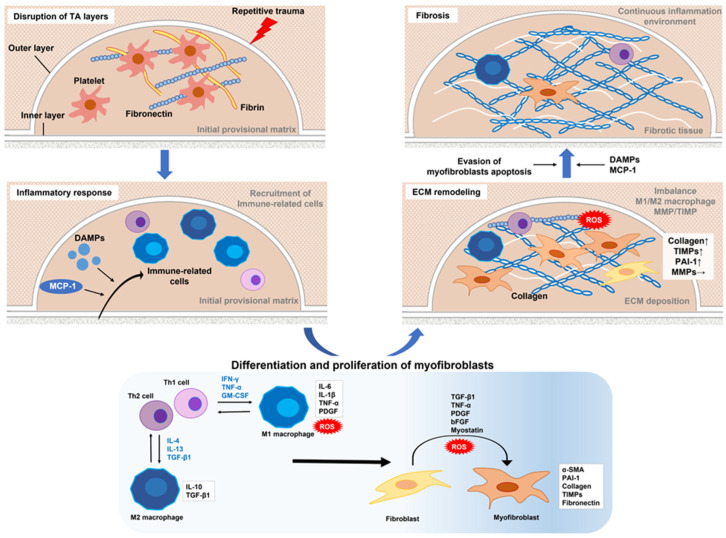
Proposed mechanism of molecular regulation of Peyronie’s disease. Repeated microtrauma disrupts the inner and outer layers of the TA. Platelets are activated and then release granules that form an early provisional matrix composed of fibrin and fibronectin. In addition, DAMPs and MCP-1 released from various cells upon tissue damage or mechanical stress promote the recruitment of inflammation-related cells to the early provisional matrix. Th1 cells produce IFN-γ, TNF-α, and GM-CSF to activate M1 macrophages, and Th2 cells produce IL-4, IL-13, and TGF-β1 to activate M2 macrophages. Activated M1 macrophages exhibit increased antigen-presenting activity and the high production of pro-inflammatory cytokines, such as IL-6, IL-1b, and TNF-α, as well as ROS. M2 macrophages are characterized by increased expressions of the anti-inflammatory cytokine IL-10 and profibrotic factor TGF-β1. An M1–M2 imbalance, expressed by excessive M2 activity, induces fibroblast activation (conversion to myofibroblasts) and promotes fibrosis. Conversion to myofibroblasts involves TGF-β1, TNF-α, IL-6, PDGF, bFGF, myostatin, and ROS. Activated myofibroblasts produce collagen and fibronectin, components of the ECM. On the other hand, the expression of MMPs that cause ECM degradation is largely unchanged by the overproduction of PAI-1 and TIMP, resulting in excessive accumulation of ECM. Increased ROS due to excessive ECM deposition, which promotes release of DAMPs due to tissue damage, and continued recruitment and stimulation of immune-related cells by MCP-1 released from myofibroblasts can lead to sustained activation of myofibroblasts. Furthermore, evasion of myofibroblast apoptosis is also critically involved in the maintenance of a chronic extracellular fiber environment. As a result, ECM continues to accumulate and is thought to promote fibrosis. TA–tunica albuginea; DAMPs–damage-associated molecular patterns; MCP-1–monocyte chemoattractant protein 1; IFN-γ–interferon gamma; TNF-α, tumor necrosis factor-alpha; GM-CSF, granulocyte-macrophage colony-stimulating factor; IL–interleukin; TGF-β1–transforming growth factor-β1; ROS–reactive oxygen species; PDGF–platelet-derived growth factor; bFGF–basic fibroblast growth factor; ECM–extracellular matrix; MMPs–matrix metalloproteinases; PAI-1–plasminogen activator inhibitor type 1; TIMP–tissue inhibitor of metalloproteinase.

**Figure 2 ijms-24-10133-f002:**
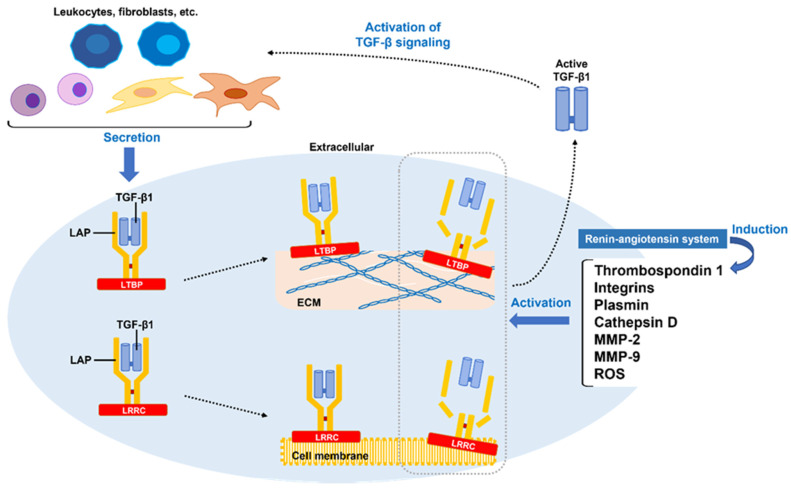
Putative mechanisms of transforming growth factor-β1 activation in the development of Peyronie’s disease. TGF-β1 is synthesized intracellularly in platelets, fibroblasts, and all leukocytes while bound being to LAP, then forms a complex with LTBP or LRRC, and is released extracellularly. Released latent TGF-β1 uses LTBP to the ECM to anchor LRRC molecules to the cell surface and maintain their inactivated state. Several factors, including integrins, thrombospondin 1, ROS, and several proteases (plasmin, cathepsin D, MMP-2, MMP-9), activate latent TGF-β1 by inducing LAP degradation, denaturation, or conformational changes. Expression of thrombospondin 1 is driven by the renin–angiotensin system. Activated TGF-β1 promotes fibrosis by further activating the TGF-β signaling pathway in macrophages and myofibroblasts. TGF-β1–transforming growth factor-β1; LAP–latency-associated polypeptide; LTBP–latent TGF-β-binding protein; LRRC–leucine-rich repeat containing; ECM–extracellular matrix; ROS–reactive oxygen species; MMP–matrix metalloproteinase.

**Figure 3 ijms-24-10133-f003:**
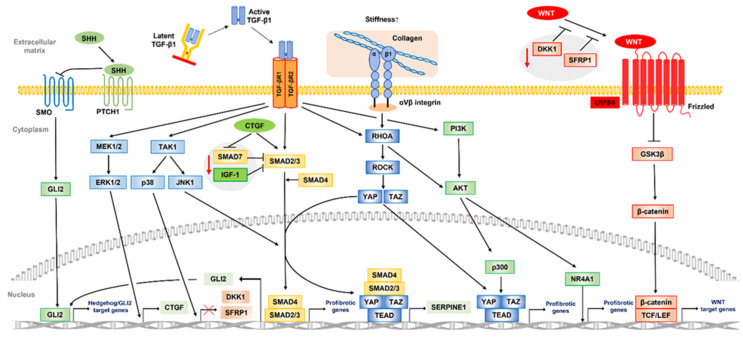
Schematic representation of core pathways involved in development of Peyronie’s disease and their interrelationships. In canonical TGF-β signaling, binding of TGF-β1 to the receptor leads to phosphorylation of SMAD2/SMAD3, triggering a cascade of responses. Phosphorylated SMAD2/SMAD3 in a complex with SMAD4 is then translocated to the nucleus, leading to the increased transcription of profibrotic genes. SMAD7, an inhibitory SMAD, is downregulated in PD and activates the TGF-β/SMAD signaling pathway. CTGF is directly induced by ERK1/2 and by non–canonical TGF-β signaling and then promotes TGF-β1 activity by decreasing SMAD7 expression and activating SMAD2 phosphorylation. IGF-1, which inhibits the transition of SMAD3 to nucleation, is also decreased in PD and contributes to the promotion of the TGF-β/SMAD pathway. TGF-β1 signaling also induces transcription of the profibrotic Hedgehog transcription factor GLI2, upregulates WNT ligands, and activates WNT/β-catenin signaling. Under activation of WNT signaling, GSK-3β loss of function prevents β-catenin phosphorylation. Subsequently, the unphosphorylated form of β-catenin accumulates in cytoplasm and enters the nucleus and then finally activates WNT target genes with the help of TCF/LEF family proteins. TGF-β1 inhibits the transcription of the WNT suppressors DKK1 and SFRP1 in a p38-dependent manner, and further activates the WNT/β-catenin signaling pathway involved in fibrosis. The profibrotic mechanism of TGF-β1-induced PI3K/AKT signaling involves the inactivation of NR4A1, which inhibits the expression of profibrotic genes. αVβ integrins activate intracellular signaling pathways, including RHOA/ROCK signaling. Subsequently, formation of F-actin stress fibers and translocation of YAP/TAZ to the nucleus are activated, promoting profibrotic gene expression via TEAD activation. Additionally, RHOA/ROCK signaling activates p300 in the nucleus through the activation of AKT, further promoting the transcription of profibrotic genes. YAP and TAZ also bind to the phosphorylated SMAD2/SMAD3 and SMAD4 complexes formed in the nucleus, and induce SERPINE1 transcription, thus leading to the excessive accumulation of ECM. TGF-β1–transforming growth factor-β1; CTGF–connective tissue growth factor; ERK1/2–extracellular signal-regulated kinase 1/2; IGF-1–insulin growth factor-1; PD–Peyronie’s disease; GLI2–GLI family zinc finger 2; WNT–wingless/int1; DKK1–dickkopf-related protein 1; SFRP1–secreted frizzled-related protein1; PI3K/AKT–phosphoinositide 3-kinase/RAC-α serine/threonine protein kinase; NR4A1–nuclear receptor subfamily 4 group A member 1; ROCK–RAS homologue gene family-associated kinase; YAP/TAZ–yes-associated protein 1/transcriptional coactivator with PDZ-binding motif; SERPINE1–serpin family E member 1; SHH–sonic hedgehog; SMO–smoothened; JNK–c-Jun N-terminal kinase; GSK3β–glycogen synthase kinase 3; TCF/LEF–T cell factor/lymphoid enhancer factor; LRP5/6–low-density lipoprotein receptor-related protein 5/6.

**Figure 4 ijms-24-10133-f004:**
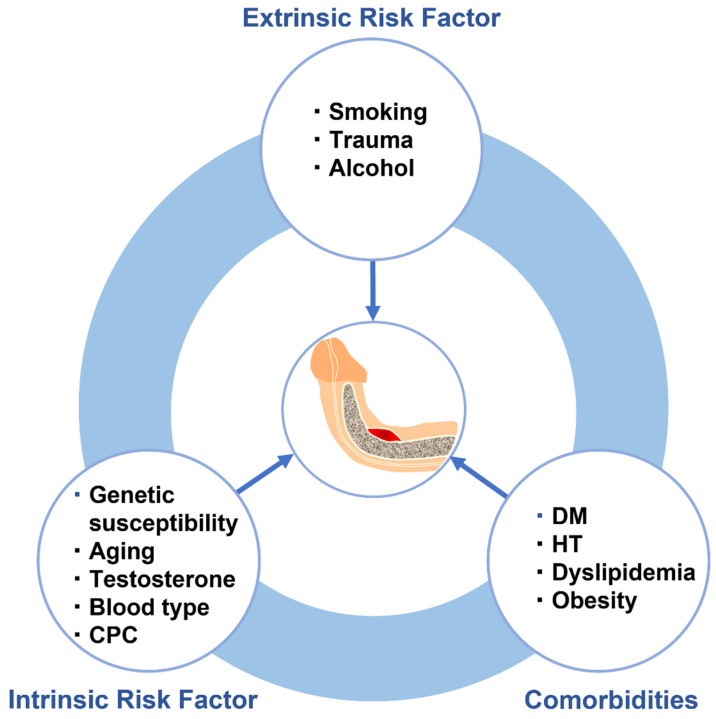
Risk factors contributing to pathogenesis of PD and their correlation diagram. A variety of comorbidities and intrinsic and extrinsic factors, both individually and in conjunction with each other, increase the risk for the development of Peyronie’s disease. DM–diabetes mellitus; HT–hypertension; CPC–congenital penile curvature.

**Table 1 ijms-24-10133-t001:** Summary of molecules associated with development of Peyronie’s disease and their roles.

Associated Molecules	Production Sources	Upregulators/Activators	Physiological Activities
MCP-1	Monocytes, macrophages, fibroblasts,	Lipopolysaccharide,	Recruitment and activation of macrophages and monocytes
	vascular endothelial cells	inflammatory cytokines	
DAMPs	Dead cells, proliferating neutrophils,	Stress, tissue injury, ROS	Recruitment and activation of immune-related cells through toll-like receptors
	macrophages, lymphocytes,		
	natural killer cells, resident cells,		
	mesenchymal stem cells		
IFN-γ	Th1 cells	IL-12	Activation of M1 macrophage polarization
TNF-α	Th1 cells, macrophages (M1)	IL-12	Activation of M1 macrophage polarization,
			promotion of differentiation of myofibroblasts
GM-CSF	Th1 cells	IL-12	Activation of M1 macrophage polarization
IL-6	Macrophages (M1)	IFN-γ, TNF-α, GM-CSF	Promotion of differentiation of myofibroblasts
TGF-β1	Platelets, fibroblasts, Th2 cells,	Th2 cells, PAI-1, integrin,	Promotion of differentiation of myofibroblasts
	macrophages (M2)	thrombospondin 1, plasmin,	
		MMP-2, -9, ROS	
PDGF	Endothelial cells, macrophages (M1),	TGF-β1	Promotion of differentiation of myofibroblasts
	platelets		
bFGF	Various cell types	Tissue injury	Promotion of differentiation of myofibroblasts
			and fibroblast mitogen
Myostatin	Skeletal muscle		Promotion of differentiation of myofibroblasts
ROS	Macrophages (M1), various cells	Hypoxia, inflammation	Promotion of differentiation of myofibroblasts through
			ROS-mediated HIF-1α stabilization
MMPs	Myofobroblasts	TGF-β1, PDGF, TNF-α, bFGF	Extracellular matrix degradation
TIMPs	Macrophages, fibroblasts, myofibroblasts	TGF-β1	Suppression of MMP activity
PAI-1	Various cell types	TGF-β1, collagen	Inhibition of plasmin-mediated MMP activation

Abbreviations: MCP-1—monocyte chemoattractant protein 1; DAMPs—damage-associated molecular patterns; IFN-γ—interferon gamma; TNF-α—tumor necrosis factor-alpha; GM-CSF—granulocyte-macrophage colony-stimulating factor; IL-6, 12—interleukin-6, 12; TGF-β1—transforming factor-beta 1; PDGF—platelet-derived growth factor; bFGF—basic fibroblast growth factor; ROS—reactive oxygen species; MMP—matrix metalloproteinase; TIMPs—tissue inhibitor of metalloproteinases; HIF-1α—hypoxia inducible factor-1 alpha; PAI-1—plasminogen activator inhibitor-1.

**Table 2 ijms-24-10133-t002:** Overview of core pathways and their roles in development of Peyronie’s disease.

Pathway	Related Factor/Pathway	Role in Fibrosis
TGF-β signaling		
Canonical	TGF-β1, TGF-βR, CTGF, SMAD2, 3, 4,	Transcriptional induction of profibrotic genes
	SMAD7 (downregulation), IGF-1 (downregulation)	(αSMA, Collagen, PAI-1, MMPs, fibronectin, GLI2, etc.)
Non-canonical		
MAPK/ERK signaling	TGF-β1, TGF-βR, MEK, ERK	Transcription induction (CTGF)
	TGF-β1, TGF-βR, TAK1, p38	Transcription repression (DKK1, SFRP1)
	TGF-β1, TGF-βR, TAK1, JNK1	Transcriptional induction of profibrotic genes
PI3K/AKT signaling	TGF-β1, TGF-βR, PI3K, AKT, p300, NR4A1	Transcriptional induction of profibrotic genes
WNT/β-catenin signaling	WNT, GSK-3β, β-catenin, TCF, LEF,	Transcriptional induction of WNT target genes
	DKK1 (downregulation), SFRP1 (downregulation)	
Hedgehog signaling	SHH, PTCH1, SMO, GLI2	Transcriptional induction of hedgehog/GLI2 target genes
	TGF-β signaling pathway (canonical)	Stimulation of fibroblast-to-myofibroblast transition
YAP/TAZ signaling	Integrins, RHOA, ROCK, YAP, TAZ	Transcriptional induction of profibrotic genes
	TGF-β signaling pathway (non-canonical)	Transcription induction (SERPINE1)

Abbreviations: TGF-β1—transforming factor-beta1; TGF-βR—TGF-β receptor; CTGF—connective tissue growth factor; IGF-1—insulin growth factor-1; αSMA—alpha-smooth muscle actin; PAI-1—plasminogen activator inhibitor type 1; MMP—matrix metalloproteinase; GLI2—GLI family zinc finger 2; ERK—extracellular signal-regulated kinase; TAK1—TGF-β-activated kinase1; DKK1—dickkopf-1; SFRP1—secreted frizzled-related protein 1; JNK—JUN N-terminal kinase; PI3K—phosphatidylinositol-3-kinase; NR4A1—nuclear receptor subfamily 4 group A member 1; WNT—wingless/int1; GSK-3β—glycogen synthase kinase 3beta; TCF—transcription factor; LEF—lymphoid enhancer-binding factor; SHH—sonic hedgehog; PTCH1—protein patched homolog 1; SMO—smoothened; RHOA—Ras homolog family member A; ROCK—Rho-associated kinase; YAP—Yes-associated protein 1; TAZ—transcriptional coactivator with PDZ-binding motif.

**Table 3 ijms-24-10133-t003:** Risk factors associated with development of Peyronie’s disease and their possible effects.

Risk Factors	Proposed Mechanisms and Pathologic Effects	Molecular Biological Changes	Reference
**Intrinsic risk factors**			
Genetics			
Chromosomal abnormalities	Familial aggregation		[158,159]
Additions deletions			
Inversions reciprocal translocations			
Single–nucleotide polymorphisms			
*TGF-β1* (rs1800471)	Activation of TGF-β pathway	TGF-β1↑	[160]
*WNT2* (rs4730775)	Activation of WNT/β-catenin pathway	WNT2↑	[161]
*WNT2* (rs4730775)	Activation of WNT/β-catenin pathway	WNT2↑	[161]
*ALMS1* (rs45501594, rs34071195,	Evasion of apoptosis	TGF-β1↑	[162,163,164]
rs41291187)			
Epigenetic modifications			
HDACs (up–regulation)	Activation of TGF-β pathway	PPARGC1A↓	[165,166]
MicroRNA-29b (down–regulation)	Regulation of expression of multiple cytokines		
	DNA methylation EMT		[167,168]
Aging	Increase in PD-related comorbidities	α1-antitrypsin↓ ROS↑	[169]
	Semi-rigid erection		[170]
	Genetic alterations, epigenetic modifications		[171]
Low testosterone level	Abnormal collagen metabolism in TA	TGF-βRII↑	[172,173,174,175,176]
ABO blood type (type O)	Promotion of inflammatory reaction	TNF-α↑ IL-6↑	[177,178,179]
	Recombination of TGF-β1R1 and ABO gene		
Congenital penile curvature	More penile trauma during intercourse		[177]
**Extrinsic risk factors**			
Smoking	Induction of profibrotic factor	CTGF↑ PAI-1↑ ROS↑	[180]
	Induction of EMT		
	Epigenetic modifications	HDACs↑	[181]
Alcohol	Increase in PD-related comorbidities		
	Induction of profibrotic factors	TIMP-1↑ SMAD3↑ MMP-9↑	[182,183]
Perineal and penile trauma	Promotion of detachment of septal fibers		
**Comorbidities**			
DM, HT, dyslipidemia, obesity	Vascular endothelial cell damage	TGF-β1↑ MMPs↑ TIMPs↑ROS↑	[17]

Abbreviations: TGF-β–tratransforming growth factor-β; WNT–Wingless/int1; ALMS–Alström syndrome; PD–Peyronie’s disease; ROS–reactive oxygen species; TGF-βR–TGF-β receptor; TNF-α–tumor necrosis factor-α; IL-6–interleukin-6; EMT–epithelial–mesenchymal transition; CTGF–connective tissue growth factor; PAI-1–plasminogen activator inhibitor type 1; HDACs–histone deacetylases; TIMP–tissue inhibitor of metalloproteinase; SIM1–SIM BHLH transcription factor 1; MMP–matrix metalloproteinase. The up and down arrows indicate up-regulation and down-regulation, respectively.

## Data Availability

Not applicable.

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
