# Peer review of "Molecular Mechanisms and Risk Factors Related to the Pathogenesis of Peyronie’s Disease"

_ijms, 2023, doi:10.3390/ijms241210133_

Round 1
Reviewer 1 Report
The current study aims to provide better understanding regarding involvement of risk factors in the molecular mechanisms associated with PD pathogenesis.
The study provides a ranking of the main andrological topics investigated in the literature, also presenting the top list of the most productive authors for each one.
The authors should be congratulated for the work and for addressing an interesting topic. Only few points warrant mentions:
Minor comments:
1. In the “Introduction” section, I would recommend to the authors to describe also the persistence of pain in the chronic phase, especially by younger patients which also experiments with the major psychological consequences, as described in PMID:36426559.
2. In the “Introduction” section, line 41, Dupuytren’s disease is not a risk factor for PD even if they are strictly associated.
3. In the “Introduction” section, line 52, I suggest to the authors to report the design of the current review (systematic, narrative, literature).
4. In section 4, “4.3.3. Dupuytren’s disease”, I recommend again to the authors to not list Dupuytren’s disease in the risk factors. Maybe would be better to report ED and Dupuytren’s disease in a separate section 5.
5. An overall English revision is required.
The Quality of the English language should be improved.
Author Response
Thank you for reviewing our manuscript, titled as shown above. Enclosed, please find a revised version as well as detailed explanations of the changes presented following. We sincerely appreciate the comments and suggestions from the reviewers, which were very helpful to improve the study..
Minor comments:
- In the “Introduction” section, I would recommend to the authors to describe also the persistence of pain in the chronic phase, especially by younger patients which also experiments with the major psychological consequences, as described in PMID:36426559.
Answer: We appreciate this important recommendation. Accordingly, text has been added to the revised Introduction section, as shown following.
“...while a recent study showed that younger patients tend to suffer more psychological and physical symptoms, as well as penile pain in the chronic phase of PD [9].”
- In the “Introduction” section, line 41, Dupuytren’s disease is not a risk factor for PD even if they are strictly associated.
Answer: Thank you for pointing this out. We have removed the statement that DD is a risk factor for PD from the text and made the following changes.
In the Introduction section, the related sentence has been changed to the following: “In addition, epidemiological studies have indicated various risk factors possibly associated with its development, such as smoking, hypertension (HT), diabetes mellitus (DM), and elderly age [10, 11].”
In Section 2.2, the related sentence has been changed to the following: “Interestingly, a recent study that used single-cell RNA sequencing (scRNA-Seq) with samples from patients with Dupuytren’s disease (DD), widely known to be associated with PD, showed that the endothelium functions to promote an immune regulatory fibroblast phenotype through PDGF signaling during fibrosis development [64].”
- In the “Introduction” section, line 52, I suggest to the authors to report the design of the current review (systematic, narrative, literature).
Answer: Thank you for this good suggestion. The following sentence is now included in the Introduction.
“The present literature review provides a summary of current findings regarding molecular mechanisms related to PD pathogenesis and also the putative roles of several risk factors including genetic associated with its development.”
- In section 4, “4.3.3. Dupuytren’s disease”, I recommend again to the authors to not list Dupuytren’s disease in the risk factors. Maybe would be better to report ED and Dupuytren’s disease in a separate section 5.
Answer: We agree with the reviewer. In the revised version, ED and DD are excluded from the discussed risk factors, and are now described separately in Section 5. Furthermore, those two factors have been removed from revised Figure 4.
- An overall English revision is required
Answer: A native speaker of English has checked the paper, including the revised portions.
Reviewer 2 Report
Dear authors,
Your review shows objectively a summary of intrinsic and extrinsic factors contributing to the formation of Peyronie's disease. Although this disease is a benign condition that shows a different clinical profile than other penis-related issues, the review focuses on their complexity and the relevance to extending their knowledge among physicians.
I believe this review is timely submitted, scientifically sound, and well-written. I only have one observation about Table 1.
Minor comments
Could you add all references supporting observations in Table 1, please?
Author Response
Thank you for reviewing our manuscript, titled as shown above. Enclosed, please find a revised version as well as detailed explanations of the changes presented following. We sincerely appreciate the comments and suggestions from the reviewers, which were very helpful to improve the study.
Minor comments
Could you add all references supporting observations in Table 1, please?
Answer: We appreciate this important recommendation. Table 1 has been revised and references related to each observation are now included.
Reviewer 3 Report
This is a good quality review on the current pathophysiology of Peyronie's disease at a molecular level.The authors recall the currently accepted mechanism of microtraumas as a probable starting point. I think that they could have discussed at the level of the molecular mechanisms the action of certain treatments used, such as extracorporeal lithotripsy, which is currently offered at the acute stage of the disease despite creating microtraumas. What is the molecular action (or not) of these treatments that could explain a beneficial effect?
Author Response
Reviewer #3
This is a good quality review on the current pathophysiology of Peyronie's disease at a molecular level. The authors recall the currently accepted mechanism of microtraumas as a probable starting point. I think that they could have discussed at the level of the molecular mechanisms the action of certain treatments used, such as extracorporeal lithotripsy, which is currently offered at the acute stage of the disease despite creating microtraumas. What is the molecular action (or not) of these treatments that could explain a beneficial effect?
Answer: We appreciate the reviewer for noting these important points and presenting the question. As a result, we consider it necessary to add more information regarding currently applied treatment methods for PD in the Introduction section. Extracorporeal shockwave therapy (ESWT) may have some efficacy, though its clinical significance is controversial at this time from the aspect of its assumed mechanism of action. The following has been added to section 2.4 in the revised version of the manuscript.
“The effect of ESWT on PD in the early phase is thought to be related to direct destruction of penile plaque by shock waves as well as plaque lysis caused by activation of macrophages induced by an inflammatory response to heat generated by the treatment [92]. However, as noted above, macrophage activation is a major factor that promotes fibrosis and the clinical significance of ESWT remains questionable. A recently published systematic review and meta-analysis shows findings indicating that ESWT may reduce plaque size, though fails to improve penile curvature or pain in PD patients [13].”
Reviewer 4 Report
The authors describe molecular mechanisms and risk factors related to PD with a narrative review.
Topic of current interest. Moderate scientific accuracy and general organization of the paper.
Some major issues are present together with several minor issues.
The English of whole paper should be improved by a English native speaker.
There are some sections in red. The authors should check the paper to remove them
Line 26: I suggest clarifying that a large percentage of patients in the acute phase do not complain of pain
I suggest to improve the quality of figures
I suggest emphasizing the limitations of the literature and therefore of the reported results
The paper is quite long and the huge number of molecules and pathways cited make it confusing. I recommend to summarize it. For this reason, in addition, I suggest to add a table with the key molecules/pathways involved in the pathogenesis of PD (reporting briefly their role)
I suggest to add a small section about the available therapies in the Introduction. These are some useful references: 10.1038/s41443-022-00651-8 (tadalafil), 10.1111/and.13527 (collagenase) , 10.1111/andr.13240 (ESWT), 10.1016/j.sxmr.2022.01.001 (modeling), 10.1016/j.jsxm.2020.07.079 (grafting), 10.23736/S2724-6051.20.03890-4 (penile prosthesis)
The English of whole paper should be improved by a English native speaker.
Author Response
Reviewer #4
The authors describe molecular mechanisms and risk factors related to PD with a narrative review. Topic of current interest. Moderate scientific accuracy and general organization of the paper. Some major issues are present together with several minor issues.
The English of whole paper should be improved by a English native speaker.
Answer: A native speaker of English has checked the paper, including the revised portions.
There are some sections in red. The authors should check the paper to remove them
Answer: Thank you for pointing this out. Sections corrected based on previous reviewers’ comments were highlighted in red. In this second revised version, those initial changes are now shown in black, while revisions to this latest version are presented in red.
Line 26: I suggest clarifying that a large percentage of patients in the acute phase do not complain of pain
Answer: Thank you for this good suggestion. The following sentence has been added to the Introduction section.
“However, not all are presented with penile pain at onset, with the incidence ranging from 20% to 70% of reported cases [1].”
I suggest to improve the quality of figures
Answer: We appreciate this important recommendation. The drawings are our own creation and it cannot be denied that the quality is poor. Asking a professional illustrator to create it was discussed, but then was not done due to financial reasons. One of the goals of this review is to make it easier for the readers to understand PD based on currently available information. We would like to use the figures in their present form and hope that the reviewer will understand the situation.
I suggest emphasizing the limitations of the literature and therefore of the reported results
Answer: Thank you for pointing this out. We believe that the study’s limitations include the subjective nature of the cited literature, which may be misleading to the readers. Another drawback is that only a summary of the various research methods is presented. The following sentence has been to the final part of the Conclusion section.
“Although the present review has several limitations, including the non-uniformity of referenced literature (mixture of clinical and basic studies) and the subjective method used for selection of those studies, it is our hope that the information presented herein will contribute to a deeper understanding of PD.”
The paper is quite long and the huge number of molecules and pathways cited make it confusing. I recommend to summarize it. For this reason, in addition, I suggest to add a table with the key molecules/pathways involved in the pathogenesis of PD (reporting briefly their role)
Answer: We appreciate this important recommendation, and agree that the many molecules and pathways discussed in this review may cause confusion for the readers. To make the information easier to understand, we have summarized key molecules associated with PD in new Table 2 and also show the main pathways in new Table 3.
I suggest to add a small section about the available therapies in the Introduction. These are some useful references: 10.1038/s41443-022-00651-8 (tadalafil), 10.1111/and.13527 (collagenase) , 10.1111/andr.13240 (ESWT), 10.1016/j.sxmr.2022.01.001 (modeling), 10.1016/j.jsxm.2020.07.079 (grafting), 10.23736/S2724-6051.20.03890-4 (penile prosthesis)
Answer: Based on the reviewer's helpful suggestion, we have added the following to the Introduction section regarding treatment methods for PD and also cite those recommended reference materials.
“A large variety of medical treatments have been suggested and are utilized for PD. Nonsurgical treatments for affected patients include oral, topical, intralesional, extracorporeal shockwave, and traction therapy. As an oral treatment option for PD, results have shown that low-dose tadalafil may slow the progression of penile curvature in patients in an acute phase [10], while intralesional therapy with collagenase clostridium histolyticum is reported to be capable of improving penile curvature [11]. Additionally, extracorporeal shockwave therapy (ESWT) used for treatment of the penis can cause plaque damage, resulting in plaque resorption in some affected patients [12, 13]. As for chronic PD, assistive devices (penile traction, vacuum devices, penile prostheses) may be effective for improving penile curvature [14, 15]. When these treatments are ineffective, surgical treatment such as a grafting procedure is often attempted to correct penile curvature [16].”
Round 2
Reviewer 4 Report
The authors changed the paper according to my suggestions
No changes required